# State-of-the-Art Design and Optimization of Strain Gauge-Type Load–Displacement Transducer for in *In Situ* Nanoindentation Systems

**DOI:** 10.3390/s25030609

**Published:** 2025-01-21

**Authors:** Duhui Lu, Jianliang Liu, Mukai Wang, Sen Gu

**Affiliations:** 1Key Laboratory of Special Equipment Safety and Energy-Saving for State Market Regulation, China Special Equipment Inspection & Research Institute, Beijing 100029, China; luduhui@csei.org.cn (D.L.); wangmukai@csei.org.cn (M.W.); 2School of Aerospace Engineering, Xi’an Jiaotong University, Xi’an 710049, China; 3School of Mechanical Engineering and Urban Rail Transit & School of Intelligent Manufacturing, Changzhou University, Changzhou 215021, China

**Keywords:** force–displacement transducer, *in situ*, scanning electron microscope, strain gauge-type

## Abstract

Force–displacement transducers are key components in *in situ* nanoindentation systems. The current existing capacitance-type transducers adopted in state-of-the-art commercial *in situ* nanoindentation systems are restricted by limited maximum ranges, and strain gauge-type transducers in the current *in situ* nanoindentation systems have the limitation of low resolution and high values of mass. In the paper, we propose a mechanical design and improvement of a low-mass strain gauge-type force–displacement transducer capable of performing high resolution *in situ* nanoindentation measurements. The transducer mainly consisted of a parallelogram-shaped flexure hinge and two strain gauges. Air resolution, *in situ* resolution, and mass experiments reported an air force resolution of 5 μN, an *in situ* force resolution of 5 μN inside a scanning electron microscope (SEM), and a mass of 6.53 g of such a strain gauge-type load–displacement transducer. Then, the transducer was assembled into a newly developed *in situ* nanoindentation–atomic force microscope (AFM) hybrid system and a newly developed *in situ* nanoindentation system. The proposed design successfully performed nanoindentation measurements inside SEM. Based on the results, the proposed strain gauge-type transducer shows great advantages compared to the current state-of-the-art transducers.

## 1. Introduction

As the miniaturization trend continues to develop in industry, functional coatings with micro- to nano-scale thickness show great application. For instance, Low-k coatings are widely used in semiconductor manufacturing to allow continued scaling of microelectronic devices [1]. Organic semiconducting coating is popular in solar batteries for improving energy conversion efficiency [2]. Titanium—(TiN, TiNCr, TiAlN) has applications in precision cutting tools for improving hardness, toughness, wear-resistance, and heat-resistance [3,4]. The wide practical applications of coatings with micro- to nano-scale thickness and their mechanical characterization [5].

An instrumented indentation testing method is widely applied for mechanical characterization of coatings with micro- to nano-scale thickness due to its simple sample preparation and instrument automation. Mechanical properties of nanostructures are remarkable different from their bulk counterparts owing to scale effects [6]. Small-scale mechanical testing (SSMT) offers many advantages and encompasses a variety of techniques of instrumented indentation testing, micro-compression, micro-bending, and micro-tensile testing [7]. Micro-compression, micro-bending, and micro-tensile testing methods require samples to be in a standard shape, which limits the possibility of analyzing a material in its original shape. Whereas, the instrumented indentation [8,9] testing method characterized materials in an original state directly using indenters with standard shape tips, which are better for coating with micro- to nano-scale thickness due to their simple sample preparation and instrument automation. Instrumented indentation testing can be performed by real-time observation utilizing high-resolution scanning electron microscopy (SEM) image feedback. SEM produces images of a sample under test by raster scanning the sample under test surface line by line with a focused beam of electrons [10]. The electrons interact with atoms in the sample, producing signals that contain information about the sample’s surface. With the advantage of a resolution of a few nanometers with a high depth focus, the indenter tip can be guided to the region of interest (ROI) of coating with micro- to nano-scale thickness before instrumented indentation [11]. Moreover, the events occurred in the surface of the sample during instrumented indentation and can be observed in real-time, which allows for the visualization and detection of a correlation of events that accompany coating deformation with the corresponding features in the force–displacement curves [12,13,14,15].

The instrumented indentation testing method continuously measures the load and displacement data of the indenter tip into the sample materials and, therefore, a force–displacement transducer is the key component in *in situ* nanoindentation systems. A number of *in situ* nanoindentation systems have been developed by companies, including Bruker [16], Nanomechanics [17], Femtotools [18], and ALEMNIS [19], as well as by academic laboratories [20]. These systems have been applied to the mechanical characterization study of coating with micro- to nano-scale thickness. Capacitance-type transducers are integrated in *in situ* nanoindentation systems from Bruker, Nanomechanics, and Femtotools. Strain gauge-type transducers are used in *in situ* nanoindentation systems from ALEMNIS to academic laboratories. The current existing capacitance-type transducers adopted in the state-of-the-art commercial *in situ* nanoindentation systems are restricted by limited maximum ranges. Strain gauge-type transducers in the current *in situ* nanoindentation systems have the limitation of low resolution and high values of mass.

In the paper, a novel strain gauge-type force–displacement transducer is proposed to perform large, measured ranges. The proposed transducer mainly consists of one parallelogram-shaped flexure hinge and two strain gauges. The state-of-the-art transducers with commercial *in situ* nanoindentation systems are introduced and their limitations are compared in Section 2. Section 3 presents the design of proposed strain gauge-type transducer and the comparison with the state-of-the-art transducers are conducted in Section 4. Section 5 describes the practical applications for proposed strain gauge-type transducer. Finally, Section 6 concludes the paper.

## 2. State-of-the-Art

The state-of-the-art force–displacement transducers used in commercial *in situ* nanoindentation systems include capacitance-type and strain gauge-type systems, as shown in Figure 1. The capacitance-type transducer measures the force-induced displacement of two parallel suspending springs as a capacity variation in two parallel capacitive electrodes, as shown in Figure 2. An indentation force applied to the suspending spring results in a change in the gaps between the electrodes, which can be measured by the change in the capacitance(1)$$C_{1}-C_{2}=\varepsilon \cdot A\cdot \left(\frac{1}{d-\Delta d}-\frac{1}{d+\Delta d}\right)$$
where *C* is the capacitance; *ε* is the permittivity of environment; *d* is the initial gap between the capacitor electrodes; Δ*d* is the change in such a gap (induced by the applied force); and *A* is the overlapping area of the capacitor electrodes. The force sensitivity *S*_cap-f_ is as follows:(2)$$S_{cap\text{-}f}=\frac{\partial V_{out}}{\partial F}=C_{CVC}\cdot \varepsilon \cdot A\cdot k\cdot \left(\frac{2}{d^{2}-\Delta d^{2}}\right)$$
where *C_CVC_* is the constant of capacitance-to-voltage converter and *k* is the stiffness of the suspending springs. According to Equation (2), the capacitance-type transducer has an inherent non-linearity and the non-linearity in voltage output relative to the force is typically larger for larger force measurement. Thus, capacitance-type transducers are used only for small force. The capacitance-type transducers with commercial *in situ* nanoindentation systems PI 85 (Bruker [16]), NANOFLIP (Nanomechanics [17]), FT-NMT04 (Femtotools [18]) have maximum measured indentation forces of 30 mN, 50 mN, and 20 mN, respectively.

The strain gauge-type force sensor measures force based on the variation in specific resistance induced by applied stress. It has a larger measurement compared to the capacitance-type transducer. The transducers with a commercial *in situ* nanoindentation system from ALEMNIS have two ranges of 0.5 N and 1.5 N. The transducer with an *in situ* nanoindentation system from academic laboratories has a maximum range of 1.5 N. However, these strain gauge-type transducers have low resolutions. They have large masses, which limit their practical application in small-scale characterization systems inside SEM.

The present work focuses on designing a strain gauge-type force–displacement transducer with a maximum measurement of 1.5 N, a better resolution and low mass.

## 3. Transducer Design

As shown in Figure 3a, the proposed transducer mainly consists of a parallelogram-shaped flexure hinge, two strain gauges and one indenter tip. Two strain gauges are glued on the upper face of the parallelogram-shaped flexure hinge, and the right end of parallelogram-shaped mechanism is fixed. When the transducer takes a measurement, two tension strain gauge resistors (RM1, RM4) and two compression resistors (RM2, RM3) form a full Wheatstone bridge circuit configuration to maximize sensitivity. The force sensitivity and displacement sensitivity of transducer are as follows:(3)$$\left\{\begin{array}{c} S_{M\text{-}For}=\frac{\Delta V_{M\text{-}out}}{\Delta F_{M}}=\frac{V_{CC}\times \left(\left|k_{M1}\cdot \varepsilon_{M1}\right|+\left|k_{M2}\cdot \varepsilon_{M2}\right|+\left|k_{M3}\cdot \varepsilon_{M3}\right|+\left|k_{M4}\cdot \varepsilon_{M4}\right|\right)}{\Delta F_{M}}\\ S_{M\text{-}Disp}=\frac{\Delta V_{M\text{-}out}}{\Delta L_{M}}=\frac{V_{CC}\times \left(\left|k_{M1}\cdot \varepsilon_{M1}\right|+\left|k_{M2}\cdot \varepsilon_{M2}\right|+\left|k_{M3}\cdot \varepsilon_{M3}\right|+\left|k_{M4}\cdot \varepsilon_{M4}\right|\right)}{\Delta L_{M}} \end{array}\right.$$
where *k_M_*_1_, *k_M_*_2_, *k_M_*_3_, *k_M_*_4_ are the factors of strain gauges; *ε_M_*_1_, *ε_M_*_2_, *ε_M_*_3_, *ε_M_*_4_ are the strain values at two clamped ends of the deformation beams; and *V_CC_* is the reference voltage of the Wheatstone bridge. The gauge factors *k_M_*_1_, *k_M_*_2_, *k_M_*_3_, *k_M_*_4_ and reference voltage of the Wheatstone bridge *V_CC_* are constants; therefore, the strain should be optimized to improve sensitivities of the proposed strain gauge-type transducer.

The strain occurred on the parallelogram-shaped flexure hinge mechanism is given by the following:(4)$$\varepsilon_{M}=\frac{3FL}{2EBT^{2}}$$
where *E*, *T*, *B*, and *L* are the elasticity modulus, thickness, width, and length of the parallelogram-shaped flexure hinge mechanism. According to Equation (4), the sensitivity depends on its material and geometric parameters, and the thickness *T* is the most important influencing factor. Firstly, the length *L* and width *B* of parallelogram-shaped flexure hinge mechanisms are determined as 30.5 mm and 6.4 mm. Then, applying Equation (4), the strain with different thicknesses could be calculated. The finite-element method (FEM) was also used to analyze the relationship between the thickness and strain of the parallelogram-shaped flexure hinge mechanism. The material of the parallelogram mechanism is Beryllium copper with a Young’s modulus of 11.5 × 10^4^ MPa, and a Poisson ratio of 0.3. A fixed boundary is applied on the right end during the FEM static structural analysis.

Table 1 shows the strain values of parallelogram-shaped flexure hinge mechanisms with different thicknesses. It can be found that the calculated values are close to the simulated values, and the difference between the calculated and simulated values increases with the increase in the thickness. This indicates the strain can be simulated by the FEM simulation method. Considering a larger strain at clamped ends and process difficulty, the thickness of the parallelogram-shaped mechanism is selected as 0.15 mm. Figure 3b displays the photograph of the proposed strain gauge-type transducer.

In Figure 4a, the results show the stress distribution of the principal stress. It can be found that the maximum value of principal stress is 184.65 MPa when the applied normal load is 1.5 N. For the Beryllium copper material, the ultimate tensile strength is 460 MPa. Therefore, the parallelogram-shaped flexure hinge mechanism is safe under a maximum load of 1.5 N. Figure 4b shows the relationship between the vertical deflection and the normal load. The goodness of the linear fitting is R^2^ = 0.99786. It can be observed that the deflection increases linearly with the increasing of the normal load. This indicates that the stiffness of the parallelogram-shaped mechanism is changeless in the measured range of 1.5 N.

## 4. Results

In this section, we focus on conducting experiments to study and compare the performances of proposed strain gauge-type transducers with those of state-of-the-art strain gauge-type transducers, as shown in Figure 1b. The experiments include air resolution, *in situ* resolution and mass. Transducer resolution experiments were carried out by setting the load to zero and holding it for a specified time period. This measures the noise due to internal electronic noise and external mechanical vibrations, and the resolution is acquired by quantifying the noise level. In our test, air resolution and *in situ* resolution experiments are performed. To quantify the noise level, it is necessary to clarify the term “resolution” that is used in this work, where we follow the ISO 5725 Standard [21].

Figure 5, Figure 6 and Figure 7 show the air force resolution, proposed transducer, and the state-of-the-art strain gauge-type transducers. In our tests, the three transducers were placed on the same vibration-isolation system to ensure the same level of mechanical vibration. The three transducers were connected to one customized readout circuit to ensure the same level of internal electronic noise. Figure 5a displays the force noise data of the proposed transducer within 600 s. Figure 5b quantifies the noise level, and the force resolution of the proposed transducer is found to be 5 μN. Figure 6a displays the force noise data of the state-of-the-art strain gauge-type transducer with a maximum measurement range of 1.5 N within 600 s. Figure 6b quantifies the noise level, and the force resolution is found to be 36.12 μN. Figure 7a displays the force noise data of the state-of-the-art strain gauge-type transducer with a maximum measurement range of 0.5 N within 600 s. Figure 7b quantifies the noise level, and the force resolution of the transducer is found to be 6.72 μN.

Figure 8 shows the *in situ* force resolution of a proposed strain gauge-type transducer. In our testing, the transducer was placed inside the vacuum chamber of the SEM for a few hours prior to the experiment to ensure that the transducer had reached thermal equilibrium with the SEM. Figure 8a shows the force noise data of the transducer inside the SEM within 120 s. Figure 8b quantifies the noise level, and the force resolution of the proposed transducer is found to be 5 μN.

Figure 9 shows the comparison of results of three strain gauge-type transducers. The proposed strain gauge-type transducer has a very low mass of 6.53 g, compared to the large mass of 94.37 g of the state-of-the-art strain gauge-type transducer with a maximum measurement range of 1.5 N, and a large mass of 91.38 g of the state-of-the-art strain gauge-type transducer with a maximum measurement range of 0.5 N.

Table 2 presents the performance comparison of the proposed strain gauge-type transducer with its counterparts. As compared, the proposed strain gauge-type transducer achieves a large maximum measured range, better resolution, and low mass simultaneously. The superior performance of the proposed strain gauge-type transducer is justified in Section 5.

## 5. Application

In this section, we focus on presenting the practical applications of proposed strain gauge-type transducers. The current existing *in situ* nanoindentation systems are limited by the lack of capability to quantify the residual imprints in real time after indentation. The quantification of residual imprints after indentation is beneficial to the study of deformation and failure mechanism analysis. SEM only produces 2D images of the test sample surface without depth information. AFM is a widely used method of mapping and measuring topography with atomic spatial resolution for any sample under test. In recent years, many scholars have studied the mechanical mechanism of nanostructures by quantifying residual imprints after indentation using the AFM technique. However, transferring a coating of micro- to nano-scale thickness back and forth while switching between AFM and SEM may cause damage and contamination to the coating, and relocating the same region of interest on the coating surface at the nanometer scale after transfer is a laborious and challenging task. By combining the AFM imaging-in-SEM technique (Figure 10a), the residual imprints of the nanoindentation can be quantified in real time without the transferring process, such that fresh information for the study of deformation and failure mechanism analysis can be acquired promptly. However, SEM chambers pose large challenges for nanoindentation–AFM hybrid system design, and limited load capability of the SEM specimen stage. The proposed strain gauge-type transducer has a significantly low mass and a better *in situ* resolution of 5 μN, which was successfully assembled in a newly developed *in situ* nanoindentation-atomic force microscope (AFM) hybrid system inside SEM, as shown in Figure 9a. Figure 10b shows one nanoindentation load–depth curve of coating TiAlSiN with the nanoindentation–AFM hybrid system.

As shown in Figure 11a, the proposed strain gauge-type transducer was successfully assembled in a newly developed *in situ* nanoindentation system inside SEM. Compared with the state-of-the-art commercial *in situ* nanoindentation system (Figure 1a), this system exhibits a larger indentation range of 1.5 N. Figure 11b shows the nanoindentation load–depth curve of target force 5 mN of coating TiAlSiN.

Figure 12 displays the linear relationship between the force and displacement of the proposed transducer under the target forces of 700 μN, 850 μN, and 1100 μN. The better resolution ensures quantitatively measuring the elastic modulus of micro- to nano-scale materials.

## 6. Conclusions

A mechanical design and improvement of a strain gauge-type force–displacement transducer with a maximum measurement of 1.5 N was proposed in this research. The transducer mainly consists of a parallelogram-shaped flexure hinge and two strain gauges. To compare the strain gauge-type transducers with the state-of-the-art commercial *in situ* nanoindentation systems, we conducted the experiment of air resolution, *in situ* resolution, and mass testing. Air resolution experiments reported a force resolution of 5 μN, compared to 6.72 μN and 36.12 μN of the commercial *in situ* nanoindentation systems. The *in situ* resolution experiment reported a force resolution of 5 μN, compared to 8 μN and 30 μN of the commercial one. Mass experiments reported a low value of 6.83 g, compared to 94.37 g and 91.38 g of the commercial one. The proposed strain gauge-type transducer was successfully assembled in a newly developed *in situ* nanoindentation-atomic force microscope (AFM) hybrid system and a newly developed *in situ* nanoindentation system for indentation measurement. The practical applications of the proposed strain gauge-type transducer show great advantages compared with those of the state-of-the-art commercial *in situ* nanoindentation systems.

## Figures and Tables

**Figure 1 sensors-25-00609-f001:**
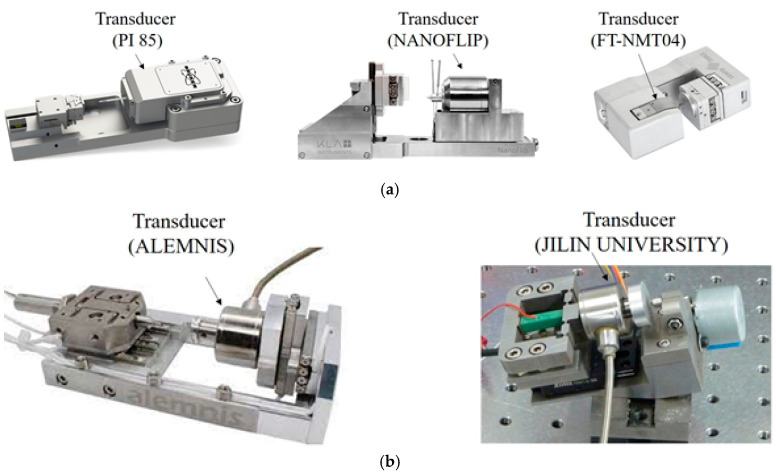
Force–displacement transducers in an *in situ* nanoindentation system. (**a**) Capacitance-type. (**b**) Strain gauge-type.

**Figure 2 sensors-25-00609-f002:**
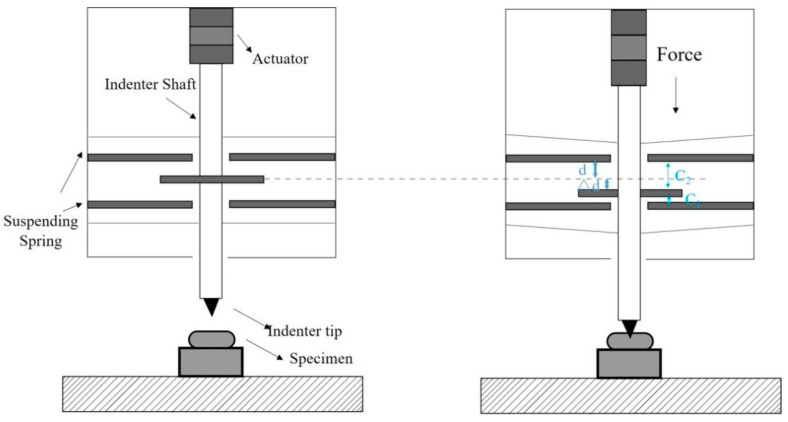
Working principle of capacitance-type transducer.

**Figure 3 sensors-25-00609-f003:**
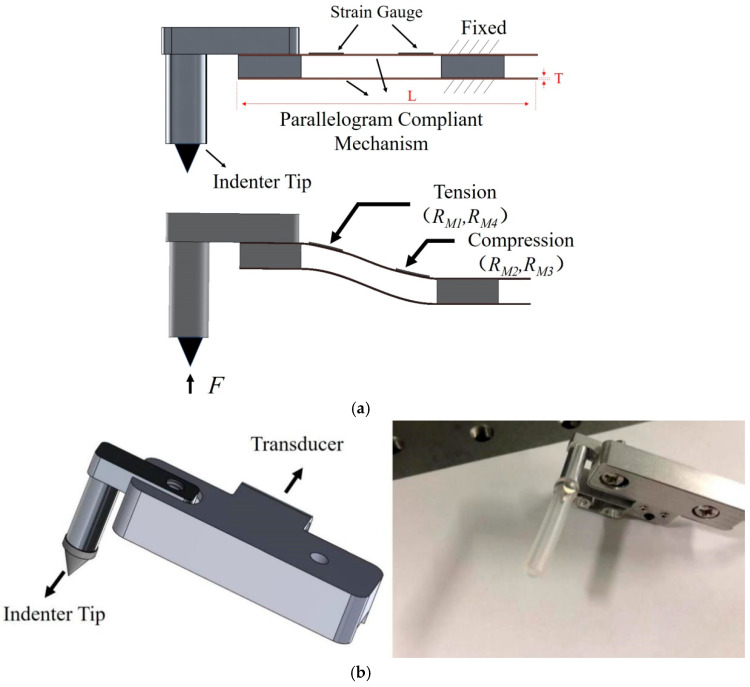
Proposed strain gauge-type transducer. (**a**) Working principle. (**b**) Photograph.

**Figure 4 sensors-25-00609-f004:**
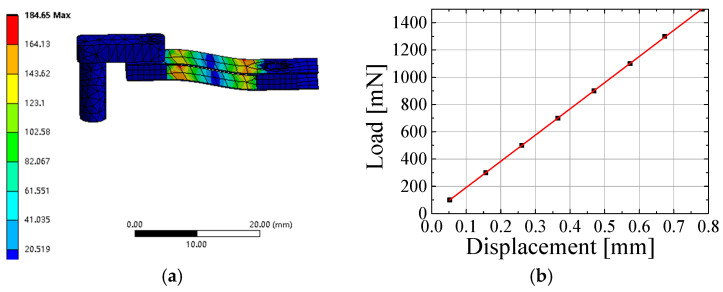
FEM simulation results. (**a**) Stress distribution of the principal (Mpa). (**b**) The relationship between the applied load and the deflection where R^2^ = 0.99786.

**Figure 5 sensors-25-00609-f005:**
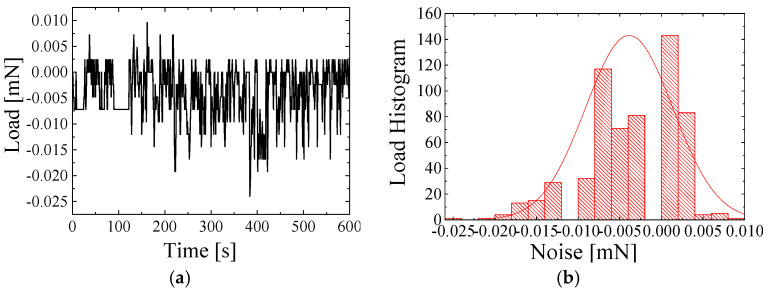
Proposed strain gauge-type transducer resolution. (**a**) Noise data. (**b**) The corresponding noise histogram.

**Figure 6 sensors-25-00609-f006:**
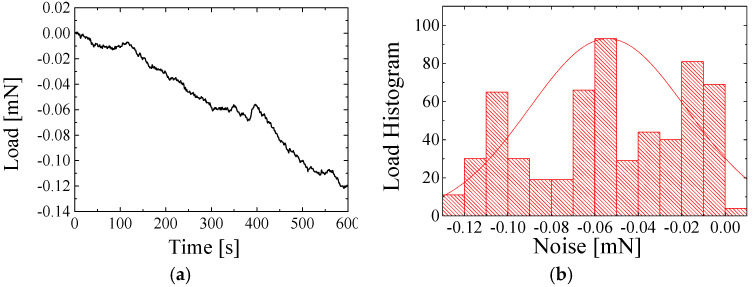
The state-of-the-art strain gauge-type transducer with a maximum measured range of 1.5 N. (**a**) Noise data. (**b**) The corresponding noise histogram.

**Figure 7 sensors-25-00609-f007:**
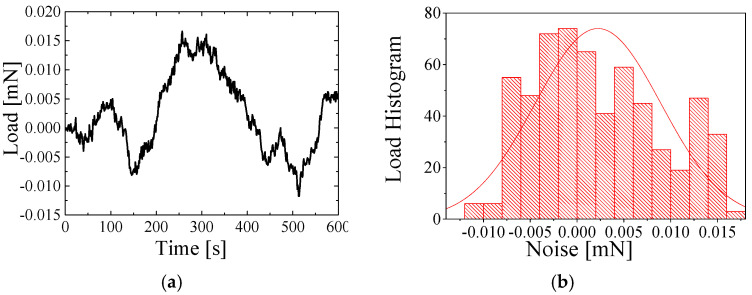
The state-of-the-art strain gauge-type transducer with a maximum measured range of 0.5 N. (**a**) Noise data. (**b**) The corresponding noise histogram.

**Figure 8 sensors-25-00609-f008:**
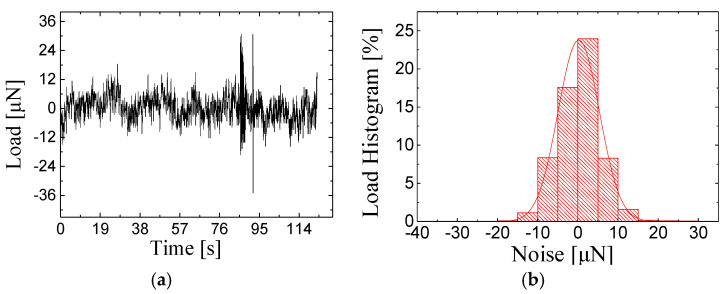
Proposed strain gauge-type transducer force resolution inside SEM. (**a**) Noise data. (**b**) The corresponding noise histogram.

**Figure 9 sensors-25-00609-f009:**
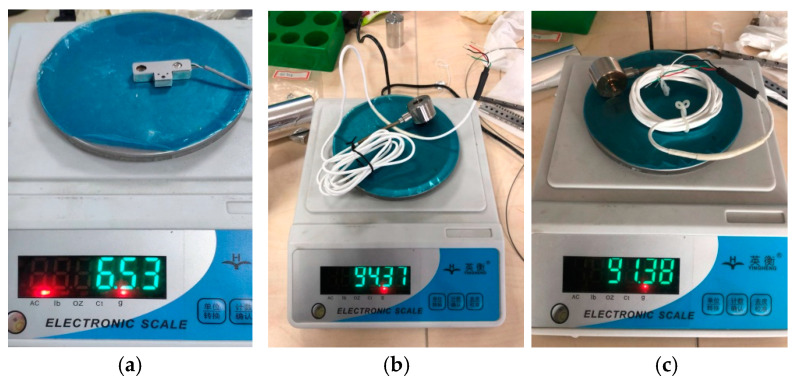
Mass comparison experiments. (**a**) Proposed strain gauge-type transducer. (**b**) State-of-the-art strain gauge-type transducer with a maximum measurement range of 1.5 N. (**c**) State-of-the-art strain gauge-type transducer with a maximum measurement range of 0.5 N.

**Figure 10 sensors-25-00609-f010:**
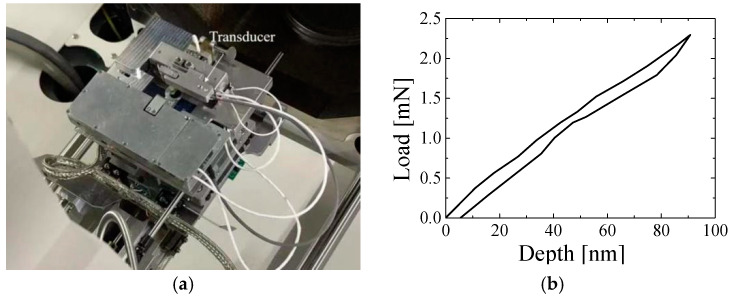
(**a**) *In situ* nanoindentation–AFM hybrid system inside SEM. (**b**) Load–depth curve.

**Figure 11 sensors-25-00609-f011:**
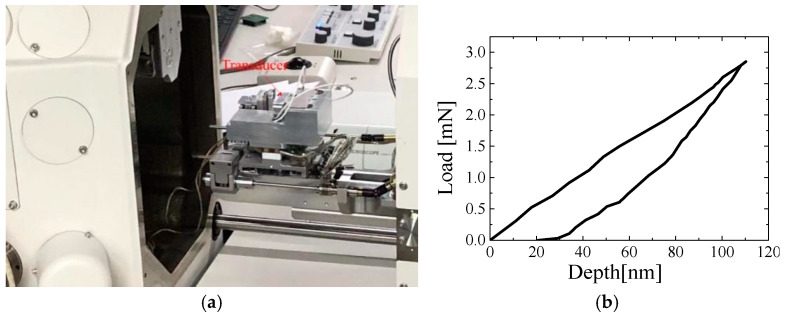
(**a**) *In situ* nanoindentation system inside SEM. Location 1. (**b**) Load–depth curve.

**Figure 12 sensors-25-00609-f012:**
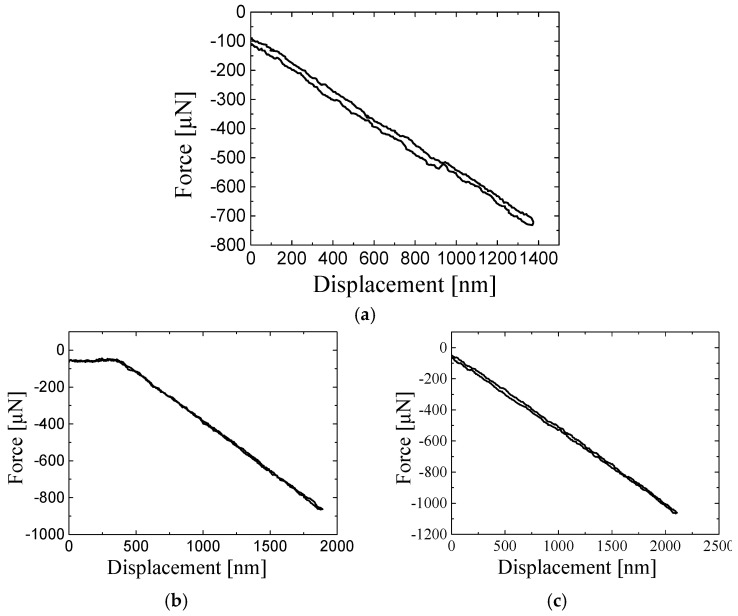
The relationship between force and displacement under different amounts of target force. (**a**) 700 μN. (**b**) 850 μN. (**c**) 1100 μN.

**Table 1 sensors-25-00609-t001:** The strain values of the parallelogram-shaped flexure hinge mechanism.

Thickness (mm)	0.05	0.1	0.15	0.2	0.25	0.3
Simulated	1.2058 × 10^−4^	3.0131 × 10^−5^	1.346 × 10^−5^	7.6476 × 10^−6^	4.946 × 10^−6^	3.4707 × 10^−6^
Calculated	1.173913 × 10^−4^	2.93478 × 10^−5^	1.30434 × 10^−5^	7.33695 × 10^−6^	4.69565 × 10^−6^	3.26086 × 10^−6^
Relative error (%)	2.6	2.6	3.1	4.1	5.1	6.1

**Table 2 sensors-25-00609-t002:** Performance comparisons.

	Maximum Indentation Force	Resolution	Mass
Ref. [16]	30 mN	0.4 μN	-
Ref. [17]	50 mN	3 nN	-
Ref. [18]	20 mN	0.5 μN	-
Ref. [19]	1.5 N	8 μN	94.37 g
Ref. [20]	1.5 N	30 μN	94.37 g
This paper	1.5 N	5 μN	6.53 g

## Data Availability

The datasets generated during and/or analyzed during the current study are available from the corresponding author on reasonable request.

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
