# Peer review of "State-of-the-Art Design and Optimization of Strain Gauge-Type Load–Displacement Transducer for in In Situ Nanoindentation Systems"

_sensors, 2025, doi:10.3390/s25030609_

Round 1
Reviewer 1 Report
Comments and Suggestions for Authors
This paper presents a novel strain gauge-type force-displacement transducer that can perform large measured ranges. The subject is interesting and can add knowledge to the force-displacement transducer development. The introduction and the subsequent section contain the subject to be studied with current and adequate references (around 40% are within the last five years). The paper is well structured, but several typos and grammatical errors were found. The methodology is usually used for this kind of study. The results are very interesting and promising. The discussion of the results is robust and based on the earlier findings. The conclusions are based on the results and discussion, showing that the mechanical design and improvement of a strain gauge-type force-displacement transducer with a maximum measurement of 1.5 N was successfully developed in this research. The paper can be accepted after the authors carefully review the grammar.
Author Response
The detailed reply to the questions please see the attachment.

Reviewer 2 Report
Comments and Suggestions for Authors
The work lacks significantly in quality and value. It is a poor and uncoherent presentation, hard to follow and reproduce. The language is awkward and repetitive. I recommend rejection.
There are too many comments and questions at essentially any line of the manuscript. Few examples:
-The title is misleading, the manuscript is about strain gauge-type transducers not capacitance-type force-displacement transducers
-There are no details on how the analytical formula (4) is applied to a rather non-ideal geometry to include the contributions of the fixed pads and how their dimensions were factored in.
-How the strain gauge factors in equation (3) were determined?
-Some of the figures are included to fill in the space, example Figure 9 showing the weighting of the transducers on scales.
-A nephogram is a picture of clouds not stress
-The comparison of the noise level between the fabricated transducer and the commercial ones lacks details to determined that this comparison is indeed fair and relevant.
-There is description of a so-called nanoindentation-AFM hybrid system. Where is the AFM? The curves presented in Figures 10 and 11 look very noisy and it is not clear what is the main conclusion out of these measurements.
-It is also unclear what the tests shown in Figure 2 want to convey: the response is noisy, the forces are negative, and the left part of the curves in Figure 12b are horizontal for some reason. There are no explanations on how these curves were acquired and on what materials.
-There are no counterpart measurements on the commercial transducers to have a comparison with what is shown in Figures 10 to 12.
Comments on the Quality of English LanguageThe language is awkward and repetitive.
Author Response

(The authors gave the same response as above.)

Reviewer 3 Report
Comments and Suggestions for Authors
This paper presents the State-of-the-Art Design and Optimization of Capacitance-Type Force-Displacement Transducer for in in situ Nanoindentation Systems. Additionally, they suggest a mechanical design and enhancement of a low-mass strain gauge-type force-displacement transducer capable of conducting high-resolution in situ nanoindentation measurements. Nonetheless, some modifications are necessary prior to possible publishing.
Authors must improve the structure of the manuscript. It is necessary to focus on the figures as well.
The author must include the problem statement in the introduction section. Additionally, the author fails to provide sufficient details regarding the experimental setup and methodologies employed.
Authors must include the error margins or standard deviation in their results.
Authors need to add more discussion about In-Situ performance.
Authors must improve the introduction and discussion sections.
Author must furnish further details regarding the setup, calibration, and validation techniques to substantiate the validity of their results.
Author Response

(The authors gave the same response as above.)
